# Utilizing Aerobic Capacity Data for EDSS Score Estimation in Multiple Sclerosis: A Machine Learning Approach

**DOI:** 10.3390/diagnostics14121249

**Published:** 2024-06-13

**Authors:** Seda Arslan Tuncer, Cagla Danacı, Furkan Bilek, Caner Feyzi Demir, Taner Tuncer

**Affiliations:** 1Software Engineering, Faculty of Engineering, Firat University, 23119 Elazığ, Turkey; satuncerl@firat.edu.tr (S.A.T.); cdanaci@firat.edu.tr (C.D.); 2Department of Software Engineering, Faculty of Technology, Sivas Republic University, 58070 Sivas, Turkey; 3Department of Gerontology, Fethiye Faculty of Health Sciences, Muğla Sıtkı Koçman University, 48000 Muğla, Turkey; furkanbilek@mu.edu.tr; 4Department of Neurology, School of Medicine, Fırat University, 23119 Elazig, Turkey; cfdemir@firat.edu.tr; 5Computer Engineering, Faculty of Engineering, Firat University, 23119 Elazığ, Turkey

**Keywords:** aerobic capacity, Expanded Disability Status Scale, gradient boosting, machine learning, multiple sclerosis

## Abstract

The Expanded Disability Status Scale (EDSS) is the most popular method to assess disease progression and treatment effectiveness in patients with multiple sclerosis (PwMS). One of the main problems with the EDSS method is that different results can be determined by different physicians for the same patient. In this case, it is necessary to produce autonomous solutions that will increase the reliability of the EDSS, which has a decision-making role. This study proposes a machine learning approach to predict EDSS scores using aerobic capacity data from PwMS. The primary goal is to reduce potential complications resulting from incorrect scoring procedures. Cardiovascular and aerobic capacity parameters of individuals, including aerobic capacity, ventilation, respiratory frequency, heart rate, average oxygen density, load, and energy expenditure, were evaluated. These parameters were given as input to CatBoost, gradient boosting (GBM), extreme gradient boosting (XGBoost), and decision tree (DT) machine learning methods. The most significant EDSS results were determined with the XGBoost algorithm. Mean absolute error, root mean square error, mean square error, mean absolute percent error, and R square values were obtained as 0.26, 0.4, 0.26, 16, and 0.68, respectively. The XGBoost based machine learning technique was shown to be effective in predicting EDSS based on aerobic capacity and cardiovascular data in PwMS.

## 1. Introduction

Multiple sclerosis (MS) is a central nervous system disease that affects quality of life and is frequently seen in young and middle-aged people [1]. The clinical complexity of MS causes a wide range of symptoms in persons with MS (PwMS) [1,2]. Many tools have been developed in addition to clinical studies to define the severity of MS [3]. One of the most widely used tools is the Expanded Disability Status Scale (EDSS). The EDSS is a scale used to determine the progress of disease in PwMS and to evaluate the effectiveness of clinical interventions. The EDSS ranges from 0 to 10 in 0.5-point steps, based on eight separate functional system (FS) scales. Lower EDSS scores are more dependent on physical examination, while higher EDSS scores (>EDSS 6) are more dependent on ambulation. Although the EDSS contains many uncertainties, it is still accepted as the standard for measuring disability in PwMS [4,5].

Although the EDSS has been widely used for nearly 30 years, the reliability and validity of the scale is strongly dependent on the neurologist performing the neurological examination. One of the main problems with EDSS is its lack of reliability, as different neurologists may produce different results for the same patients in a series of neurological examinations [5]. Although an expert system has been developed to semi-automatically evaluate EDSS, scores of functional systems evaluated by neurologists are still required as entry parameters for this expert system [6]. Therefore, a more objective measure to measure disability in MS is required, to overcome the limitations of the EDSS.

EDSS scores increase in parallel with the decrease in the functional capacity of the individual. However, a decrease in aerobic capacity is expected in individuals with reduced functional capacity [7]. Previous studies in PwMS reported a weak to moderate correlation (range −0.25 to −0.58) between aerobic capacity and EDSS [2]. Evaluation of an individual’s functional capacity may vary from physician to physician. On the other hand, evaluation of aerobic capacity as an objective parameter reflecting functional capacity can eliminate individual differences. It has been shown that aerobic capacity is associated with functional neuronal plasticity in PwMS [8]. Neuroplasticity can be beneficial in overcoming relapses and resisting MS progression. It has been confirmed that physical exercise supports neuroplasticity, so it is not surprising that aerobic capacity is associated with functional performance, strength, fatigue, and cognitive functions in PwMS [9,10]. Increased levels of physical activity have been shown to modulate microglial activation in the central nervous system (CNS) and increase cortical thickness in MS [11,12]. For these reasons, aerobic capacity can be an important indicator for representing disabilities caused by neurodegeneration in the CNS for PwMS. This is because exercise or physical activity, which affects physical capacity, has neuroprotective properties. Additionally, the evaluation of an individual’s functional capacity can vary from one physician to another when using the EDSS assessment. On the other hand, assessing aerobic capacity as an objective parameter reflecting functional capacity can eliminate individual differences.

Artificial intelligence and machine learning-based studies that will help diagnose MS or make predictions about its progression are becoming increasingly important. The implementation of machine learning in MS has so far mainly been used to classify participants according to different disease stages [13,14]. DT-based algorithms compared to logistic regression and support vector machines were used to predict secondary progressive multiple sclerosis (SPMS) disability progression. Variables included EDSS, multiple sclerosis functional compound scores, T2 lesion volume, brain parenchymal fraction, disease duration, age, and gender. In their current form, the models developed in the study were not found to be clinically useful in predicting an individual’s disease course. However, SPMS disability progression was best predicted by non-parametric machine learning [15]. However, to identify clinically isolated syndrome patients who have converted to MS using machine learning, a focus has been placed on lesion features in magnetic resonance imaging (MRI), particularly features that describe shape and brightness. Conversion or non-conversion was predicted correctly in 71 patients based on shape features derived from computer-assisted manual segmentation masks (84.5% accuracy) [16]. There are few studies in the literature on EDSS score estimation using machine learning methods [17,18]. This study focused on generalizable and high-accuracy EDSS score prediction with machine learning methods, using PwMS patient data with EDSS scores between 0 and 5. The aim of the study was to provide a tool that can be used in the clinical management of MS patient by providing a more accurate and reliable prediction of EDSS scores with machine learning methods. In this context, using machine learning techniques, the performance of the model was tested with various algorithms, focusing on the generalizability of the results. The models used aim to provide reliable results on real world data.

### Contribution and Paper Organization

This study proposes a machine learning approach to estimate EDSS scores using aerobic capacity data of PwMS. The contributions of the study to the literature can be summarized as follows.

-A machine learning approach to predict EDSS score using the aerobic capacity data of multiple sclerosis (MS) patients.-Use of an automatic decision support system in EDSS evaluation-Use of data based not only on classical neurological symptoms but also on physiological measurements in EDSS evaluation.

Regarding the organization of this study, the Section 1 presents the purpose and scope of the study, and a review of the studies conducted in the literature. Section 2 details the dataset used in the study, the steps followed for the development of the proposed machine learning models, and the performance evaluation criteria. Section 3 presents the experimental results obtained from the machine learning algorithms, and Section 4 provides a comprehensive analysis of the potential of the findings. Finally, in the Section 5, a summary of the study is presented, the contributions made are expressed, and recommendations for future work are made.

## 2. Materials and Methods

### 2.1. Material

A total of 106 PwMS, 17 of whom were male, with a mean age of 30.87, from Fırat University Hospital were included in the study. All measurements were carried out during the same visit. Neurological evaluations (EDSS etc.) were performed by a single neurologist. All procedures were approved by the Fırat University Non-Interventional Research Ethics Committee (Protocol no:03, Decision no:10, Date: 23 February 2023) and carried out in accordance with the relevant ethical principles. Inclusion criteria for this study were as follows: definite diagnosis of relapsing-remitting multiple sclerosis (RRMS) or SPMS according to the McDonald criteria, age between 19 and 65 years, in the range of 1.0–5.0 Neurostatus-EDSS score, able to walk 20 m independently with or without an assistive device, and have not had an attack at for least 90 days before the test [19]. Participants who had significant comorbidities affecting gait and balance (e.g., Parkinson’s, stroke, ataxia, vertigo, Alzheimer’s) were excluded from the study. Informed consent was obtained from each participant. PwMS underwent a full neurological examination by a Neurostatus-qualified neurologist from our MS center at Fırat University, to calculate EDSS scores and to exclude them.

Incremental exercise testing was used to determine the cardiopulmonary status and maximum aerobic capacity (VO_2_max) of the participants. VO_2_max was assessed using a Cosmed FitMate Pro^®^ (Cosmed, Italy). The FitMate Pro^®^ includes a turbine flow meter to measure ventilation and a galvanic fuel cell oxygen sensor to analyze the oxygen fraction in expired gases. After resting for 2 min (sitting on the ergometer), the participants performed a 3 min warm-up at 0 W on a bicycle ergometer (E200 Cosmed^®^, Italy). In the loading period, the resistance was increased by 15 W every 1 min until the participants felt exhausted. Participants maintained a cadence of 60 rpm until they reached voluntary fatigue. Participants were verbally encouraged to continue the exercise as long as possible. Minute ventilation (VePeak), heart rate (KH), and VO_2_max values were automatically recorded at the time of maximum oxygen consumption. A value range table of the data obtained from the participants is given in Table 1.

### 2.2. Method

EDSS score can be determined based on classical neurological examination findings of PwMS. In this method, a neurologist evaluates the patient’s motor, sensory, visual, and cognitive functions by applying walking speed tests, visual tests, or cognitive tests, and determines the EDSS score. Determining the EDSS score depends on the experience of doctors and is subjective.

The following method was developed to determine the EDSS score of PwMS objectively and quickly.

Step 1: In the first step, aerobic capacity data of PwMS were provided as input to the designed system.Step 2: Pre-processing steps were applied to the data.Step 3: After pre-processing, the data were evaluated as training and test data using a cross-validation technique. The prediction process was performed using the training data and prediction results were obtained.Step 4: The results were compared with the actual input values, and the performance of the prediction process was evaluated using various criteria. The process design of the study is presented in Figure 1.

#### 2.2.1. Data Processing

Based on a two-step preprocessing process, the descriptive statistics of the data given in Table 2 were evaluated and the normalization preprocess was applied by examining the statistical properties and distributions of the data together.

Step 1: Before analyzing the descriptive statistics of the data, it was examined whether null values were observed in the data. Since no null values were found in the data, no preprocessing was applied in this direction.Step 2: Descriptive statistics and data distributions of the data were examined and the normalization procedures to be applied on the data were determined.

The minimum value of the minimum parameter characteristic is given in Table 2, the average value of 25% of the values contained in the 25% parameter characteristic, the median value of the 50% parameter characteristic, the average value of 75% of the values contained in the 75% parameter characteristic, and the maximum value of the maximum parameter characteristic. The descriptive statistics examined provided information about the general distribution and central tendencies of the data and served as a reference for the normalization operations that were performed on the data. Table 2 shows that there was a difference between the minimum and maximum values of the EEPeak parameter, which is reflected in the standard deviation.

When the distribution of the EEPeak feature in Figure 2 is examined, it is seen that it exhibited a non-normal distribution, as well as a wide range and high standard deviation compared to the other features.

This feature had a higher variance compared to the others and contained more outliers. Considering the current conditions, min-max transformation, which is a linear transformation technique applied for non-normally distributed data in the range [0,1], was performed for the EEPeak parameter. Considering the outliers, min-max normalization was preferred, to make this feature more compatible with other features. Outliers are data that deviate from the general distribution of a dataset. These data have the potential to negatively affect the training process of a model. However, outliers may not only negatively affect the training process, but may also represent extreme cases of real-world problems. For this reason, outliers were analyzed and a transformation process was performed to reduce the effect of outliers instead of eliminating them.

#### 2.2.2. Prediction

Prediction is a supervised machine learning technique that relates newly given inputs to dependent output variables as a function of independent variables. In this study, prediction models were created using DT, GBM, XGBoost, and CatBoost algorithms from machine learning algorithms. The Python programming language, which includes Pandas, Numpy, and Matplotlib libraries, was used to code the algorithms. To improve the generalizability and reliability of the estimation results, a k-fold cross-validation technique was used. k was set to 5. The cross-validation technique increases the objectivity of model performance by ensuring that each data point is evenly distributed in both the training and validation sets. In this study, the dataset was divided into four equal parts. After the model had been trained, validation was performed on the test set. The first subset was used as the test set, and the remaining four subsets were used for training. This procedure was repeated five times, so that each subset was used once as a test set, and the overall performance of the model was measured by averaging the validation results of the five iterations. The method used provides an understanding of how models work on various data subsets and the generalizability of the results. The model parameters of the algorithms were determined using GridSearchCV, to determine the model hyperparameters suitable for the existing data. This method searches over a wide range of parameters and applies cross-validation techniques to ensure that the model performs in a balanced manner with the correct parameters. By determining the most appropriate parameters, the GridSearchCV method strengthens the generalization capacity of classification algorithms and provides protection against overfitting. The hyperparameter ranges to be searched with the GridSearchCV method were determined in line with the value ranges commonly used in the literature on health data and the experiences obtained in previous studies. In addition, dataset characteristics and computational capacity constraints were also taken into consideration in determining the parameter ranges. Sensitivity analysis was performed to evaluate the effect of range selection on the classifier performance. Sensitivity analysis was performed to evaluate the effect of parameter ranges on classifiers. With this analysis, performance metrics such as the accuracy, precision, recall, and F1 score of the model with different parameter combinations were evaluated. A total of 5 different combinations were tried and sensitivity analysis was performed, and the best-performing range was selected as the appropriate range. The search process was carried out with the selected suitable range and the obtained parameters were given as input to the prediction models, and their performances were evaluated using multiple criteria.

##### GBM and XGBoost

GBM is a method that works with the logic of transforming weak learners into strong learners. GBM starts by using weak learning models such as decision trees. GBM corrects the errors in previous predictions for each weak model to create a new model. Each tree focuses on correcting the errors of the previous tree. Finally, the predictions of all weak models are summed and a stronger prediction is made. XGBoost can be described as an optimized version of the GBM algorithm and provides better performance, thanks to the controlled mechanism it uses to prevent overlearning. The XGBoost algorithm has become one of the most successful algorithms used in the field of machine learning in recent years, because it has advantageous features such as preventing the overlearning problem, processing empty data, providing fast running performance, parallel computing, minimum resource usage, etc. The XGBoost algorithm performs an initial prediction for all data in the first step. In each iteration, model errors are calculated and a DT is constructed to improve the predictions based on these errors. Each generated tree is evaluated by the algorithm, and depending on the complexity, a regularization term (L1 (Lasso) and L2 (Ridge)) is added to make the model robust to overfitting. This model, which offers higher performance in terms of accuracy than other models, is also more advantageous than other methods in terms of training cost.

##### Decision Tree

DT is a supervised machine learning algorithm widely used in classification and prediction processes. Classification trees are used to predict a class label (discrete) against given inputs, whereas regression trees are used to predict a continuous dependent variable from continuous and independent inputs. DT offers advantages such as the ability to model complex datasets and to deal with missing data. DT facilitates the understanding of how model predictions are realized with the heuristic approach it offers for decision-making processes. This feature gives the DT algorithm a wide range of applications. For example, it offers effective solutions with its regression structure in customer risk–value segmentation, in many stages from diagnosis to treatment selection in the field of health and in similar application areas. The main purpose of the regression tree is to divide the data into subfields and map these subfields to predicted values with the minimum error. This model makes predictions by splitting a dataset based on certain characteristics and making decisions. The dataset is first divided into subsets according to the feature that best divides it. This process is repeated for each subset, thus creating a tree structure. Leaf nodes at the end of the tree show the prediction values. The reason why DT is widely used in classification and regression processes is that the rules are clear and understandable when creating the tree structure.

##### CatBoost

CatBoost is a gradient boosting algorithm that uses a learning model based on the gradient boosting framework. Unlike other gradient boosting algorithms, CatBoost performs operations using a sequential boosting mechanism. In other algorithms, this can introduce prediction bias into the model, as all training samples are considered after several boosting steps have been performed. CatBoost, on the other hand, allows the model to test itself on new data samples at each training iteration, making it more robust to overfitting. The symmetric decision trees used in the CatBoost model optimize the model’s memory usage and reduce training time by ensuring that splits at each level occur evenly across all leaves. Combining these strategies allows CatBoost to process datasets more efficiently and make faster and more accurate predictions.

To achieve high efficiency for created models, the hyperparameters to be used in the algorithm should be adapted to the data. Hyperparameter tuning is performed to increase model predictive power, speed up the training time of the model, and prevent over-learning. In this study, GridSearchCV, a hyperparameter selection method, was used. GridSearchCV tries all combinations in a specified range of values and evaluates the model for each combination using the cross-validation method, selecting the hyperparameters with the best performance. The parameters selected for the estimation algorithms using the GridSearchCV method are given in Table 3.

#### 2.2.3. Performance Evaluation

In this study, the performance of the estimation algorithm was evaluated using the MAE, MAPE, MSE, RMSE, and R^2^ metrics. Mathematical expressions of the performance evaluation metrics used are given in Table 4. In the equations given in Table 4, *k*: is the number of observations, *x_i_*: is the actual value, and x_i_’: is the prediction value [20].

Mean absolute error (MAE): MAE measures the average absolute difference between the true value and the predicted value. The closer the MAE value is to zero, the better the performance of the model.Mean absolute percentage error (MAPE): MAPE is defined as the percentage error between actual values and predicted values. This metric shows the percentage magnitude of the errors. Lower MAPE values indicate that the model works better.Mean square error (MSE): MSE is the average of the squares of the error between actual values and predicted values. It measures the magnitude of errors and the closer it is to zero, the better the model works [21].Root mean square error (RMSE): RMSE is the square root of MSE and measures the magnitude of the error between predicted values and actual values. The closer the RMSE value is to zero, the better the model works. A value of zero means that the model makes no errors.R^2^: R^2^, takes a value between 0 and 1, expressing the relationship between predicted values and actual values. The closer the result is to 1, the higher the performance and precision of the model [21].

These evaluation metrics were used to assess the accuracy and reliability of the model. The MAE and MAPE measures express the magnitude of errors in absolute and percentage terms, while the MSE and RMSE measure the magnitude of errors in a more sensitive way.

## 3. Experimental Results

This study was carried out based on the approach of accurately estimating the EDSS score with machine learning methods using aerobic capacity data in patients with MS. The estimation process performed was evaluated using MAE, MAPE, MSE, RMSE, and R^2^ metrics. The experimental results of the study are given in Table 5, with the evaluation metrics for each k-fold cross-validation.

When Table 5 is analyzed, it can be seen that the algorithms used within the scope of this study expressed significant results. For the MS, RMSE, and MAE parameters, it is observed that the lowest level of error, which was close to 0, was achieved by the XGBoost and CatBoost algorithms. The mean values of the results obtained from each k-fold validation of the algorithms used are given in Table 6.

When Table 6 is analyzed, it can be said that, since the value of 0.26 obtained by the extreme gradient boosting and decision tree algorithms for the MAE parameter was close to 0, the error rate was low, and the result obtained was significant. When the other algorithms are analyzed respectively, it is seen that gradient boosting and CatBoost algorithms were less successful than the other algorithms for the MAE metric. When the MAPE parameter has a value between 10 and 20, this means that a correct prediction model is created.

When the MAPE values obtained in the study are analyzed, it is seen that all algorithms were within this range and expressed meaningful results. It is seen that the XGBoost algorithm gave the most significant result, with a MAPE value of 16, in support of the MAE parameter. This value was 17.2 for the GBM algorithm and 18 for the DT algorithm, and it can be concluded that the GBM algorithm gave more meaningful results than the DT. The MSE, RMSE, and R^2^ metrics also gave similar results to the other metrics, and according to these results, it can be seen that the most successful algorithm was the XGBoost algorithm. The XGBoost algorithm was followed by GBM algorithm in terms of performance, while the DT algorithm ranked third according to the evaluation metrics. The CatBoost algorithm, on the other hand, showed a less successful performance compared to the other algorithms according to the metrics obtained. If a general conclusion can be made by analyzing Table 6, it is understood that each parameter expressed significant results within its value range and the prediction models were highly accurate.

In line with the significant results obtained from the performance metrics, the test data values and the predicted values were analyzed comparatively. Figure 3 shows the test data given to the model and the line graph of the predicted outputs against these inputs.

The prediction values given in Figure 3 support the results in Table 6. When Figure 3 is analyzed, it is observed that the prediction values produced by XGBoost and GBM models were closer to the actual values compared to the other models. The XGBoost model made predictions close to the true value at most of the sample data points, showing deviations only at some points. Similarly, the GBM model also exhibited high performance, in line with the general trend. When DT and CatBoost models are analyzed, it is observed that there were more deviations in some extreme data points compared to the other two models. In particular, when some data points were analyzed, it was observed that these two algorithms produced different predictions compared to the others.

## 4. Discussion

It is recommended to evaluate the actual walking distance in the evaluation of the EDSS [4]. However, due to time and/or logistical constraints, the walking distance is often determined according to the statements of the patients. When the data of patients other than MS patients were examined, it was observed that the maximum walking distances declared by the patients did not match the actual maximum walking distances. Considering these studies, aerobic capacity data, which are related to walking distance and reflect functional capacity in PwMS, were used in this study [22]. Thus, this aimed to eliminate individual differences in determining walking distance. As a result, when the estimated values and performance metrics in this study were examined together, it was observed that results close to the real EDSS scores were obtained, and the target determined in this direction was achieved.

Cardiopulmonary fitness is considered an important determinant of health and performance and is closely related to physical activity level and sedentary time [23]. However, VO_2_max is a sensitive measure and, intuitively, cardiopulmonary fitness is expected to decrease as disability increases in PwMS [24]. Due to extensive damage to the CNS, the disease is characterized by various sensory, motor, cerebellar, and cognitive dysfunctions. These dysfunctions limit physical activity behavior in PwMS and can subsequently lead to deconditioning [25,26]. Many cross-sectional studies in small samples have reported correlations between disability level for EDSS and VO_2_max in PwMS [27,28,29]. The slope of the correlation between VO_2_max and EDSS was determined such that a one-point increase in EDSS would decrease VO_2_max by 2.6 mL·kg^−1^·min^−1^ [2]. Physical activities that affect aerobic capacity provide neuroprotection in MS, possibly by directly affecting the brain. Several studies in healthy individuals proposed that the main factors responsible for enhancing neuroplasticity associated with improved brain function post-AE are transient increases in glutamatergic-mediated intracortical excitation and decreases in γ-aminobutyric acid (GABA)-mediated intracortical inhibition [28,29,30,31]. Aerobic capacity may be an important predictor for reflecting disability severity.

Furthermore, aerobic capacity has been reported to have a possible prophylactic effect on cardiovascular disease risk, better walking performance and improved cognitive processing speed, and structural decline of brain tissue in PwMS, while impaired aerobic capacity in healthy individuals has been reported to be associated with functional limitations that may hinder independent living [8,11,32,33]. For these reasons, aerobic capacity is considered an important physiological measure in PwMS.

It is widely known that applying the right treatment as a result of an early and accurate diagnosis prevents disability and reduces healthcare costs [34,35]. For these reasons, it is highly advantageous to utilize advanced technologies in the evaluation of aerobic capacity data to diagnose EDSS. In addition, testing walking distance in clinical practice is time-consuming and difficult to implement, and this distance is often reported by the patient, increasing the need for the development of new technology-based assistive methods. In the literature on predicting EDSS values, Alves et al. aimed to predict the EDSS scores of MS patients with machine learning algorithms using clinical notes provided by neurologists. A total of 13,766 patients’ data were used within the scope of the study, 684 of which had EDSS scores obtained from the OM1 MS Registry. They performed the prediction process using the XGBoost estimator, a machine learning model. As a result of the examinations, they determined that the Spearman R value was 0.75, the Pearson R value was 0.74, the AUC value was 0.91, the positive predictive value was 0.85, and the negative predictive value was 0.85 [17]. Yang et al. aimed to calculate the EDSS score from patients’ electronic health records using natural language processing. They considered 16,441 medical records of 4808 patients who received care at an MS outpatient clinic in Canada. As a result of their study, they stated that they achieved results in accordance with their goals in the combined keyword model and stated that the study could be automated to extract clinically relevant information from unstructured notes [18]. The difference between the proposed study from previous ones is that it predicts the EDSS score with machine learning methods and using aerobic capacity data. In this study, the aim was to objectively obtain the EDDs score by analyzing aerobic capacity data (VO_2_max, VePeak, RfPeak, HRPeak, FeO_2_Peak, LoadPeak, EEPeak) in PwMS. Thus, this eliminated individual differences in determining walking distance. In addition, it increased the reliability of the EDSS and prevented the problem of the differences of opinion among physicians in EDSS score estimation. Many studies in the current literature focused on estimating the EDSS score by considering clinical data or patient statements. Estimates may be inconsistent due to individual differences, and these approaches were often based on subjective assessments of patients. Furthermore, because clinical data are often limited and time-dependent, they may be insufficient to make long-term and large-scale predictions. In our study, EDSS scores were predicted by machine learning techniques using reproducible and objective biometric measures such as aerobic capacity data. It is thought that the proposed method will enable more consistent and reliable results to be obtained in clinical applications, by providing high accuracy and reliability to support traditional methods based on subjective evaluations.

This study has several limitations. Firstly, since the study included only RRMS or SPMS patients with an EDSS between 1 and 5, the results cannot be generalized to all individuals with MS. The EDSS (Expanded Disability Status Scale) is used to assess disease progression based on the physical abilities of MS patients. The EDSS scale range varies from 0 (normal examination) to 10 (death). The scale range used in this study covered the mild and moderate stages of the disease. This limitation restricts the applicability of the results to individuals with advanced disease. To overcome this limitation, a wider EDSS scale range should be included in the study, and the scope of the study should be expanded.

Secondly, although the results of the study were quite satisfactory, it is thought that higher accuracy values would be obtained by including an equal number of PwMS for each EDSS value. It is thought that a balanced sample set would increase the generalization and accuracy of the results, by ensuring equal representation of the EDSS group. It is thought that the applicability of the study findings to a wider disease population could be increased by providing a balanced distribution for each EDSS value, by addressing the unbalanced sampling constraint in future studies.

Another limitation is that, although other diseases that could affect the aerobic capacity of the participants were excluded, the individual differences and lifestyle patterns of the participants were ignored. Considering that this study is pioneering research, there is a need for further studies that consider the individual characteristics of the participants in detail. Lastly, artificial intelligence models require high-quality, accurate, and abundant data. The dataset used in this study was collected by expert physicians in the field, reflecting a high accuracy. However, the number of data points could be increased. It is believed that increasing the number of data points would enhance the performance of the prediction model.

## 5. Conclusions

This study provided an automated decision support system modeled by machine learning methods to determine the EDSS score based on aerobic capacity data in PwMS. In addition to reducing potential complications from incorrect scoring procedures, this allows performing a differential assessment of factors with the potential to cause disease progression. According to this study, the best EDDS score prediction performance was obtained using DT. When the estimated values and performance metrics were examined together, it was observed that results close to the real EDSS scores were obtained, and the target determined in this direction was achieved. The performance of the method showed that the EDSS score, which is difficult to calculate objectively clinically, can be determined quickly and objectively and can be applied in clinical practice. It is expected that this study will lead to the standardization of EDSS score estimation, assist physicians in clinical trials, and be used as a clinical training material.

## Figures and Tables

**Figure 1 diagnostics-14-01249-f001:**
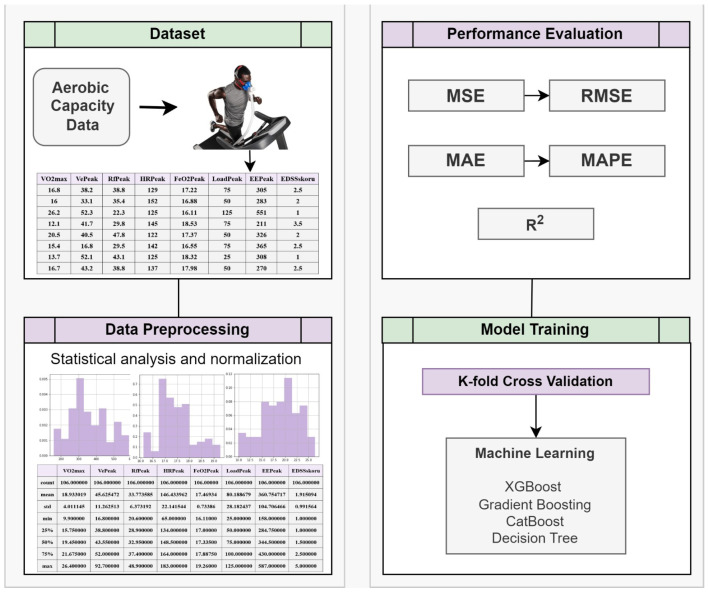
The process design of the study.

**Figure 2 diagnostics-14-01249-f002:**
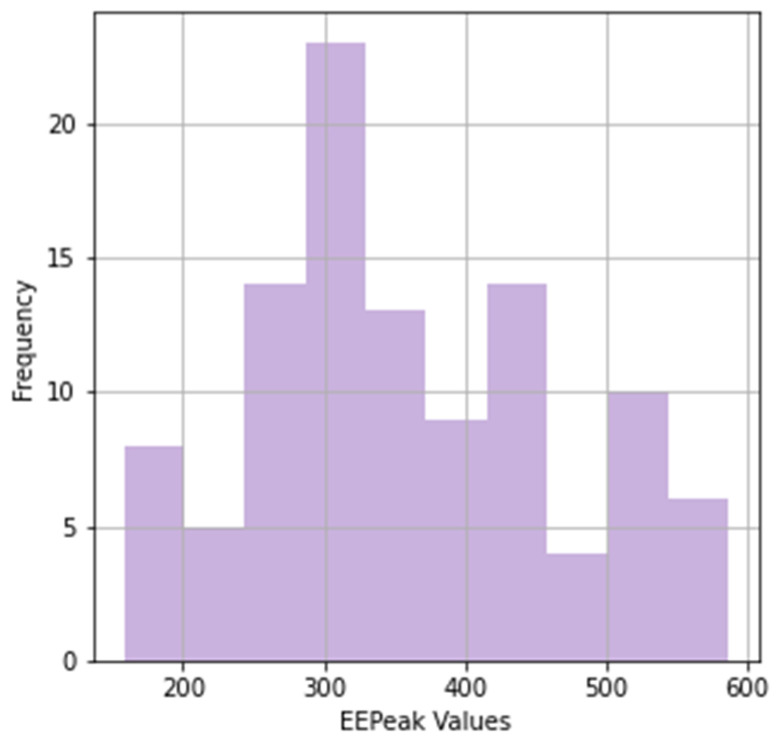
EEPeak distribution.

**Figure 3 diagnostics-14-01249-f003:**
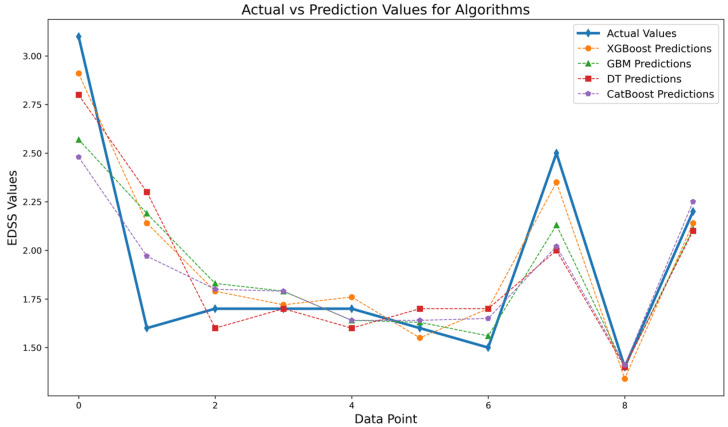
Actual and prediction value comparison chart.

**Table 1 diagnostics-14-01249-t001:** Parameter value range.

Parameter	Min Value	Max Value	Mean ± Standard Deviation
VO_2_max (mL/kg/min)	9.9	26.4	18.76 ± 4.12
VePeak (L/min)	16.8	92.7	46.31 ± 12.57
RfPeak (b/min)	20.6	48.9	34.16 ± 6.54
HRPeak (bpm)	65	183	141.39 ± 29.67
FeO_2_Peak (%)	16.11	19.26	17.34 ± 0.71
LoadPeak (Watt)	25	125	76.51 ± 25.90
EEPeak (kcal/hour)	158	587	343.39 ± 101.36
Parameter	Min Value	Max Value	Median EDSS
EDSS	1	5	1.5

VO_2_: aerobic capacity, Ve: ventilation, Rf: respiratory frequency, HR: heart rate, FeO_2_: average oxygen density, EE: energy expenditure, EDSS: Expanded Disability Status Scale.

**Table 2 diagnostics-14-01249-t002:** Descriptive statistics of the data.

	VO_2_max	VePeak	RfPeak	HRPeak	FeO_2_Peak	LoadPeak	EEPeak	EDSS
count	106.000000	106.000000	106.000000	106.000000	106.00000	106.000000	106.000000	106.000000
mean	18.933019	45.625472	33.773585	146.433962	17.46934	80.188679	360.754717	1.915094
std	4.011145	11.262513	6.373192	22.141544	0.73386	28.182437	104.706466	0.991564
min	9.900000	16.800000	20.600000	65.000000	16.11000	25.000000	158.000000	1.000000
25%	15.750000	38.800000	28.900000	134.000000	17.00000	50.000000	284.750000	1.000000
50%	19.450000	43.550000	32.950000	148.500000	17.33500	75.000000	344.500000	1.500000
75%	21.675000	52.000000	37.400000	164.000000	17.88750	100.000000	430.000000	2.500000
max	26.400000	92.700000	48.900000	183.000000	19.26000	125.000000	587.000000	5.000000

VO_2_: aerobic capacity, Ve: ventilation, Rf: respiratory frequency, HR: heart rate, FeO_2_: average oxygen density, EE: energy expenditure, EDSS: expanded disability status scale.

**Table 3 diagnostics-14-01249-t003:** Parameter values for prediction algorithms.

Paremeter	Value Range	XGBoost	GBM	DT	CatBoost
colsample_bytree	0.4, 0.5, 0.6, 0.9, 1	0.5	0.6	-	-
n_estimators	100, 200, 500	100	100	-	200
max_depth	2, 3, 4, 5, 6	4	3	5	6
learning_rate	0.1, 0.01, 0.5	0.5	0.1	0.5	0.01
min_samples_split	2, 3, 4, 5, 6	-	3	2	-

XGBoost: extreme gradient boosting, GBM: gradient boosting, DT: decision trees.

**Table 4 diagnostics-14-01249-t004:** Performance evaluation metrics [20,21].

	Performance Evaluation Metrics				
	MAE	MAPE	MSE	RMSE	R^2^
Mathematical Expression	$\frac{1}{k}\sum_{i=1}^{k}\;\left|x_{i}-x_{i}^{\prime }\right|$	$\frac{100}{k}\sum_{i=1}^{k}\;\left|\frac{x_{i}-x_{i}^{\prime }}{x_{i}}\right|$	$\frac{1}{k}\sum_{i=1}^{k}\;{\left(x_{i}-x_{i}^{\prime }\right)}^{2}$	$\sqrt{\frac{1}{k}\sum_{i=1}^{k}\;{\left(x_{i}-x_{i}^{\prime }\right)}^{2}}$	$\frac{\sum (x-\bar{x})(y-\bar{y})}{\sum (x-\bar{x}{)}^{2}\sum (y-\bar{y}{)}^{2}}$

**Table 5 diagnostics-14-01249-t005:** General performance evaluation analysis results.

Metrics	XGBoost					DT					GBM					CatBoost				
K-Fold	1	2	3	4	5	1	2	3	4	5	1	2	3	4	5	1	2	3	4	5
MSE	0.1	0.3	0.2	0.2	0.4	0.4	0.6	0.4	0.6	0.2	0.1	0.2	0.1	0.2	0.5	0.1	0.3	0.2	0.2	0.5
RMSE	0.3	0.5	0.4	0.4	0.6	0.6	0.7	0.6	0.6	0.5	0.4	0.5	0.7	0.5	0.7	0.4	0.5	0.5	0.5	0.7
MAE	0.2	0.3	0.3	0.2	0.3	0.3	0.3	0.3	0.2	0.2	0.3	0.3	0.3	0.2	0.3	0.3	0.3	0.3	0.3	0.4
MAPE	18	16	18	11	17	22	13	24	12	19	21	15	16	15	19	17	19	16	18	25
R^2^	0.8	0.5	0.8	0.5	0.8	0.6	0.7	0.8	0.7	0.9	0.8	0.6	0.8	0.4	0.7	0.8	0.5	0.6	0.7	0.7

XGBoost: extreme gradient boosting, GBM: gradient boosting, DT: decision trees.

**Table 6 diagnostics-14-01249-t006:** Average scores from performance reviews.

Avarage Performance Evaluation Metrics	XGBoost	DT	GBM	CatBoost
MSE	0.2	0.4	0.2	0.2
RMSE	0.4	0.6	0.56	0.65
MAE	0.26	0.26	0.3	0.32
MAPE	16	18	17.2	19
R2	0.68	0.74	0.66	0.66

XGBoost: extreme gradient boosting, GBM: gradient boosting, DT: decision trees.

## Data Availability

Data are contained within the article and Appendix A.

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
