# Peer review of "Utilizing Aerobic Capacity Data for EDSS Score Estimation in Multiple Sclerosis: A Machine Learning Approach"

_diagnostics, 2024, doi:10.3390/diagnostics14121249_

Round 1
Reviewer 1 Report (New Reviewer)
Comments and Suggestions for Authors
1. The paper faces inconsistency in the technical terminology usage. For example, the terms like "aerobic capacity data" and "EDSS score" are used interchangeably with "VO2max" and "RRMS or SPMS patients," which is confusing. Clarify and standardize these terms.
2. The study analysis lacks depth in explaining the certain important areas, such as the preprocessing steps for the aerobic capacity data and details of how the ML algorithms were trained and tested.
3. While the author acknowledges several limitations, such as the inclusion criteria and dataset size in the study. Provide more comprehensive discussion of these limitations and elaborate on how these limitations might have influenced the results and provide suggestions for future research to address them.
4. While the paper describes the use of GridSearchCV for hyperparameter tuning, it lacks discussion on the rationale behind the selection of specific parameter ranges and the impact of these choices on model performance. Provide insights into how the parameter ranges were determined. Were any sensitivity analyses conducted to assess their influence?
5. The paper does not delve deeper in to the handling outliers in the data, discuss the potential limitations arising from outlier presence.
6. Check the consistency in the design and formatting of tables 3,4,5,6 in the paper to maintain the uniformity. Especially pay attention on the factors like font size, style and improve visual quality of the figures presented in the manuscript.
7. The descriptions of the hyperparameter tuning process and the performance evaluation metrics are convoluted and difficult to follow. Clearer explanations are needed to understand the key aspects of the study.
8. The paper fails to provide sufficient contextualization of the study within the existing literature. There is a lack of discussion on how the findings contribute to or align with previous research in the field.
9. The paper should address the objective of the study and clearly articulate the research implications of the findings.
10. Provide clear presentation of the results using appropriate tables, figures and captions to understand the findings effectively. The current manuscript tables and figures are not explained clearly, for instances how the methods demonstrate these result value and implication behind them. Additionally, potential limitation for those values are not provided.
Comments on the Quality of English LanguageAs part of the peer review process, I have carefully reviewed the manuscript and identified areas where improvements in language clarity, grammar, and style may enhance the overall readability and professionalism of the manuscript.
Author Response
Revision Letter
Dear Editor and Reviewer,
We would like to sincerely thank the reviewers and editor for providing us with constructive and insightful feedback. The paper has been revised again and improved in the light of your remarks. The comments from the editor and each reviewer have been addressed below. Also, in this revision, some sections have been modified to address the reviewers’ comments, with the changing parts highlighted in yellow shadowed text through the manuscript.
Sincerely yours,
The authors.
Reviewer:
Comment 1: The paper faces inconsistency in the technical terminology usage. For example, the terms like "aerobic capacity data" and "EDSS score" are used interchangeably with "VO2max" and "RRMS or SPMS patients," which is confusing. Clarify and standardize these terms.
Response 1: Dear reviewer, we would like to express our gratitude for your valuable comment. We have corrected all inconsistencies in the technical terminology. However, we were unable to identify any errors in the terms SPMS and RRMS. Furthermore, we did not abbreviate the term "aerobic capacity" while abbreviating the maximum aerobic capacity as "VO2max" in the text. We have rechecked the entire article with this logic and have made the necessary corrections.
Comment 2: The study analysis lacks depth in explaining the certain important areas, such as the preprocessing steps for the aerobic capacity data and details of how the ML algorithms were trained and tested.
Response 2: Respected reviewer; thank you very much for your valuable comments. In line with your suggestion, data preprocessing under 2.2.1 Data Preprocessing and training and testing processes under 2.2.2 Prediction have been detailed and added to the study. The additions are as follows.
|
2.2.1 Data Preprocessing Previous Version |
2.2.1 Data Preprocessing Update Version |
|
Within the scope of the study, the descriptive statistics given in Table 2 of the data were evaluated and necessary pre-processes were applied on the data. The minimum value of the feature with the minimum parameter property given in Table 2, the mean value of 25% of the values contained in the 25% parameter feature, the median value of the 50% parameter feature, the mean value of 75% of the values contained in the 75% parameter feature, represents the largest value that the maximum parameter property has. Table 2 shows that there is a difference between the minimum and maximum values of the EEPeak parameter and this is reflected in the standard deviation value.
|
Within the scope of the study, descriptive statistics of the data given in Table 2 were evaluated and normalization pre-process was applied by examining the statistical properties and distributions of the data together. Before analyzing the descriptive statistics of the data, it was examined whether null values were observed in the data. Since there were no null values in the data, no pre-processing was applied in this direction. The minimum value of the feature with the minimum parameter property given in Table 2, the mean value of 25% of the values contained in the 25% parameter feature, the median value of the 50% parameter feature, the mean value of 75% of the values contained in the 75% parameter feature, represents the largest value that the maximum parameter property has. The descriptive statistics examined provide information about the general distribution and central tendencies of the data and serve as a reference for normalization operations that can be performed on the data. Table 2 shows that there is a difference between the minimum and maximum values of the EEPeak parameter and this is reflected in the standard deviation value.
|
|
2.2.2 Prediction Previous Version |
2.2.2 Prediction Update Version |
|
Prediction is a supervised machine learning technique that relates newly given inputs to dependent output variables as a function of independent variables. In this study, prediction models were created using DT, GBM, XGBoost and CatBoost algorithms from machine learning algorithms. Python programming language, which includes Pandas, Numpy and Matplotlib libraries, was used to code the algorithms. To improve the generalizability and reliability of the estimation results, a k-fold cross-validation technique is used. k is set to 5. The cross-validation technique increases the objectivity of the model performance by ensuring that each data point is evenly distributed in both the training and validation set. In this study, the data set was divided into four equal parts. Each sub-set is used as a test set and the rest of the sub-sets are used as training sets. The method used provides an understanding of how the models work on various data subsets and generalizability of the results. |
Prediction is a supervised machine learning technique that relates newly given inputs to dependent output variables as a function of independent variables. In this study, prediction models were created using DT, GBM, XGBoost and CatBoost algorithms from machine learning algorithms. Python programming language, which includes Pandas, Numpy and Matplotlib libraries, was used to code the algorithms. To improve the generalizability and reliability of the estimation results, a k-fold cross-validation technique is used. k is set to 5. The cross-validation technique increases the objectivity of the model performance by ensuring that each data point is evenly distributed in both the training and validation set. In this study, the data set was divided into four equal parts. After the model was trained, validation was performed on the test set. The first subset was used as the test set and the remaining four subsets were used for training. This procedure was repeated five times so that each subset was used once as a test set, and the overall performance of the model was measured by averaging the validation results of the five iterations. |
Comment 3: While the author acknowledges several limitations, such as the inclusion criteria and dataset size in the study. Provide more comprehensive discussion of these limitations and elaborate on how these limitations might have influenced the results and provide suggestions for future research to address them.
Response 3: Respected reviewer; thank you very much for your valuable comments. The following adjustments were made under Discussion 4 regarding the limitations of the study, and elaborations and suggestions for future research were included.
|
Previous Version |
Update Version |
|
This study has several limitations. First, the results cannot be generalized to all individuals with MS, as the study included only RRMS or SPMS patients with an EDSS of 1 to 5.
|
This study has several limitations. Firstly, since the study included only RRMS or SPMS patients with EDSS between 1 and 5, the results cannot be generalised to all individuals with MS. The EDSS (Expanded Disability Status Scale) is used to assess disease progression based on the physical abilities of MS patients. The EDSS scale range varies from 0 (normal examination) to 10 (death). The scale range used in the study covers the mild and moderate stages of the disease. This limitation restricts the applicability of the results on individuals with advanced disease. In order to overcome this limitation, a wider EDSS scale range should be included in the study and the scope of the study should be expanded.
|
|
Previous Version |
Update Version |
|
Secondly, although the results of the study are quite satisfactory, it is thought that higher accuracy values will be obtained by including an equal number of PwMS for each EDSS value.
|
Secondly, although the results of the study are quite satisfactory, it is thought that higher accuracy values would be obtained by including an equal number of PwMS for each EDSS value. It is thought that a balanced sample set will increase the generalization and accuracy capabilities of the results by ensuring equal representation of the EDSS group. It is thought that the applicability of the study findings to a wider disease population can be increased by providing a balanced distribution for each EDSS value by addressing the unbalanced sampling constraint in future studies. |
Comment 4: While the paper describes the use of GridSearchCV for hyperparameter tuning, it lacks discussion on the rationale behind the selection of specific parameter ranges and the impact of these choices on model performance. Provide insights into how the parameter ranges were determined. Were any sensitivity analyses conducted to assess their influence?
Response 4: Respected reviewer; thank you very much for your valuable comments. While determining the parameter ranges with the GridSearchCV method, the value ranges commonly used in the literature on health data and the experience gained from previous studies were utilized. Sensitivity analysis was performed to evaluate the effect of range selection on classifier performance. With this analysis, performance metrics such as accuracy, precision, recall and F1 score of the model with different parameter combinations were evaluated. The search process was performed by selecting the most appropriate range and the obtained parameters were given as input to the prediction models and their performance was evaluated using multiple criteria. In line with your suggestion, under 2.2.2 Prediction, the process is detailed and how the search range is determined is analyzed. The arrangement is as follows:
|
Previous Version |
Update Version |
|
The obtained parameters were given as input to the prediction models, and their performance was evaluated using multiple criteria. |
The hyperparameter ranges to be searched with the GridSearchCV method were determined in line with the value ranges commonly used in the literature on health data and the experience gained from previous studies. Sensitivity analysis was performed to evaluate the effect of the range selections on the classifier performance. With this analysis, performance metrics such as accuracy, precision, recall and F1 score of the model with different parameter combinations were evaluated. The search process was performed by selecting the most appropriate interval and the obtained parameters were given as input to the prediction models and their performance was evaluated using multiple criteria. |
Comment 5: The paper does not delve deeper in to the handling outliers in the data, discuss the potential limitations arising from outlier presence.
Response 5: Respected reviewer; thank you very much for your valuable comments. In line with your suggestion, under the heading 2.2.1 Data Preprocessing, the potential limitations caused by the presence of outliers were discussed and necessary arrangements were made. The revisions are given below.
|
Previous Version |
Update Version |
|
This feature has a higher variance compared to the others and especially contains outliers. Considering the current conditions, min-max transformation, which is a linear transformation technique applied for non-normally distributed data in the range [0-1], was performed for the EEPeak parameter. Considering the outliers, min-max normalization was preferred to make this feature more compatible with other features. |
This feature has a higher variance compared to the others and especially contains outliers. Considering the current conditions, min-max transformation, which is a linear transformation technique applied for non-normally distributed data in the range [0-1], was performed for the EEPeak parameter. Considering the outliers, min-max normalization was preferred to make this feature more compatible with other features. Outliers are data that deviate from the general distribution of the data set. These data have the potential to negatively affect the training process of the model. However, outliers may not only negatively affect the training process, but may also represent extreme cases of real-world problems. For this reason, outliers were analyzed and a transformation process was performed to reduce the effect of outliers instead of eliminating them
|
Comment 6: Check the consistency in the design and formatting of tables 3,4,5,6 in the paper to maintain the uniformity. Especially pay attention on the factors like font size, style and improve visual quality of the figures presented in the manuscript.
Response 6: Respected reviewer; thank you very much for your valuable comments. Tables 3-6 have been formatted by considering font sizes and table styles. Figures have been reorganized with improved visual quality.
Comment 7: The descriptions of the hyperparameter tuning process and the performance evaluation metrics are convoluted and difficult to follow. Clearer explanations are needed to understand the key aspects of the study.
Response 7: Respected reviewer; thank you very much for your valuable comments. The hyperparameter tuning process 2.2.2 Prediction section has been updated in GridSearchCV to improve detail and clarity. The descriptions of the performance evaluation metrics have been edited to make them more understandable.
|
Previous Version |
Update Version |
|
In this study, the performance of the estimation algorithm was evaluated using the MAE, MAPE, MSE, RMSE, and R2 metrics. Mathematical expressions of performance evaluation metrics used are given in Table 4. In the equations given in Table 4, k: is the number of observations, xi : is the actual value, xi' : is the prediction value [20]. MAE is a parameter that measures the absolute difference between the predicted value and the true value. The closer the MAE value is to zero, the higher the model performance. MAPE is a parameter that allows to represent the error between the predicted values and the actual values as a percentage. MSE is a measure of the size of the error between two continuous variables. This parameter, which always takes a positive value between 0 and infinity, indicates that the model performs better as it approaches zero [21]. The RMSE serves to measure the magnitude of the error between the predicted values and the actual values. While the RMSE value is close to zero, the model performance increases, while this value is zero indicates that the model works without error. R2, takes a value between 0-1, expressing the relationship between the predicted values and the actual values. The closer the result is to 1, the higher the performance and precision of the model [21].
|
In this study, the performance of the estimation algorithm was evaluated using the MAE, MAPE, MSE, RMSE, and R2 metrics. Mathematical expressions of performance evaluation metrics used are given in Table 4. In the equations given in Table 4, k: is the number of observations, xi :is the actual value, xi' : is the prediction value [20]. •Mean Absolute Error (MAE): MAE measures the average absolute difference between the true value and the predicted value. The closer the MAE value is to zero, the better the performance of the model. •Mean Absolute Percentage Error (MAPE): MAPE is defined as the percentage error between actual values and predicted values. This metric shows the per-centage magnitude of the errors. Lower MAPE values indicate that the model works better. •Mean Square Error (MSE): MSE is the average of the squares of the error be-tween the actual values and the predicted values. It measures the magnitude of the errors and the closer it is to zero, the better the model works [21]. •Root Mean Square Error (RMSE): RMSE is the square root of MSE and measures the magnitude of the error between predicted values and actual values. The closer the RMSE value is to zero, the better the model works. A value of zero means that the model makes no errors. •R2: R2, takes a value between 0-1, expressing the relationship between the predicted values and the actual values. The closer the result is to 1, the higher the performance and precision of the model [21]. The evaluation metrics were used to assess the accuracy and reliability of the model. The MAE and MAPE measures express the magnitude of errors in absolute and percentage terms, while the MSE and RMSE measures the magnitude of errors in a more sensitive way. |
Comment 8: The paper fails to provide sufficient contextualization of the study within the existing literature. There is a lack of discussion on how the findings contribute to or align with previous research in the field.
Response 8: Respected reviewer; thank you very much for your valuable comments. Added “Physical activities that affect aerobic capacity provide neuroprotection in MS, possibly by directly affecting the brain . several studies in healthy individuals have proposed that the main factors responsible for enhancing neuroplasticity associated with improved brain function post-AE are the transient increases in glutamatergic-mediated intracortical excitation and decreases in γ-aminobutyric acid (GABA)–mediated intracortical inhibition. Aerobic capacity may be an important predictor to reflect disability severity.” in the second paragraph of the discussion section. Thank you.
Comment 9: The paper should address the objective of the study and clearly articulate the research implications of the findings. (Hocam bu kısımda bulguları açıkça ifade edin demiÅŸ o kısımda takıldım sadece)
Response 9: Respected reviewer; thank you very much for your valuable comments. The purpose of the study is included in the “This study focuses on generalizable and high-accuracy EDSS score prediction with machine learning methods using PwMS patient data with EDSS scores between 0-5. The aim of the study is to provide a tool that can be used in the clinical management of MS patients by providing a more accurate and reliable prediction of EDSS scores with machine learning methods. In this context, using machine learning techniques, the performance of the model was tested with various algorithms and focused on the generalizability of the results. The models used are aimed to provide reliable results on real world data.” paragraph in the Introduction title, and an explanation to achieve the purpose of the study is provided in the introductory paragraph of the study.
Comment 10: Provide clear presentation of the results using appropriate tables, figures and captions to understand the findings effectively. The current manuscript tables and figures are not explained clearly, for instances how the methods demonstrate these result value and implication behind them. Additionally, potential limitation for those values are not provided. (Hocam bu noktada da birkaç düzenleme gerçekleÅŸtirdim ama yeterli mi bilmiyorum)
Response 10: Respected reviewer; thank you very much for your valuable comments. Taking your suggestion into consideration, some figure and table titles have been updated and more understandable expressions have been included. Some of the revised titles are as follows:
|
Previous Version |
Update Version |
|
“Table 6. Average performance evaluation results” |
“Table 6. Average scores from performance reviews” |
|
“Table 5. Performance evaluation results”
|
“Table 5. General Performance Evaluation Analysis Results”
|
Figure 1 has been reorganized and updated in the document to improve the clarity of the study.
Reviewer 2 Report (New Reviewer)
Comments and Suggestions for Authors
The article is devoted to a fairly relevant topic. In general, issues of both diagnostics and treatment support based on the analysis of anaerobic loads, analysis of cardiometry, specific spikes and characteristic dependencies in the work of the heart are in demand now and methods of working with big data are effective for them.
However, there are several comments to the article that may improve its acceptance. So, for an unprepared reader, the article uses an excessive number of abbreviations in a number of paragraphs, which complicate the perception of the text. Yes, their decoding is given at the first mention, but the number is very large, so it is recommended to give the full names again from time to time to remind the reader of their meaning.
Also, in the discussion section, in the reviewer’s opinion, sufficient attention is not paid to comparison with analogues of the presented methods, and this is an important point for assessing the effectiveness of the presented technology.
However, in general, the article can be recommended for publication.
Author Response
Revision Letter
Dear Editor and Reviewer,
We would like to sincerely thank the reviewers and editor for providing us with constructive and insightful feedback. The paper has been revised again and improved in the light of your remarks. The comments from the editor and each reviewer have been addressed below. Also, in this revision, some sections have been modified to address the reviewers’ comments, with the changing parts highlighted in yellow shadowed text through the manuscript.
Sincerely yours,
The authors.
Reviewer:
Comment 1: However, there are several comments to the article that may improve its acceptance. So, for an unprepared reader, the article uses an excessive number of abbreviations in a number of paragraphs, which complicate the perception of the text. Yes, their decoding is given at the first mention, but the number is very large, so it is recommended to give the full names again from time to time to remind the reader of their meaning.
Response 1: Respected reviewer; thank you very much for your valuable comments. Taking your suggestions into consideration, the full names of some abbreviations have been included at certain points in accordance with the flow of the study, thus contributing to comprehensibility.
Comment 2: Also, in the discussion section, in the reviewer’s opinion, sufficient attention is not paid to comparison with analogues of the presented methods, and this is an important point for assessing the effectiveness of the presented technology.
Response 2: Respected reviewer; thank you very much for your valuable comments. Taking your suggestion into consideration, the difference of the study with analogue methods is pointed out under the heading 4.Discussion: ‘Many studies in the current literature have focused on estimating the EDSS score by considering clinical data or patient statements. Estimates may be inconsistent due to individual differences and these approaches are often based on subjective assessments of patients. Furthermore, because clinical data are often limited and time-dependent, they may be insufficient to make long-term and large-scale predictions. In our study, EDSS scores are predicted by machine learning techniques using reproducible and objective biometric measures such as aerobic capacity data. It is thought that the proposed method will enable more consistent and reliable results to be obtained in clinical applications by providing high accuracy and reliability to support traditional methods based on subjective evaluations.’ It is given with the paragraph.
Round 2
Reviewer 1 Report (New Reviewer)
Comments and Suggestions for Authors
I appreciate the author for the revisions attempted to address the previous comments, but there are still some areas that may need further refinement for clarity. The following things need to be refined for further acceptance.
1. The revision related to the choice of parameter selection could be improved by providing more clear details on how parameter ranges were determined and what sensitivity analyses were conducted.
2. Need careful formatting throughout the paper for organizing the information in more structured manner.
3. The response adequately addresses this by mentioning updates made to figure and table titles. However, it could be improved by providing more detailed explanations of the results and their implications.
In general, the responses could be further refined by providing more explicit explanations, organizing the information in a more structured manner, and ensuring clarity throughout the manuscript.
Comments on the Quality of English LanguageMinor editing and English mistranslation corrections are needed.
Author Response
Revision Letter
Dear Editor and Reviewer,
We would like to sincerely thank the reviewers and editor for providing us with constructive and insightful feedback. The paper has been revised again and improved in the light of your remarks. The comments from the editor and each reviewer have been addressed below. Also, in this revision, some sections have been modified to address the reviewers’ comments, with the changing parts highlighted in yellow-shadowed text throughout the manuscript.
Sincerely yours,
The authors.
Reviewer:
Comment 1: The revision related to the choice of parameter selection could be improved by providing more clear details on how parameter ranges were determined and what sensitivity analyses were conducted.
Response 1: Dear reviewer, we would like to express our gratitude for your valuable comment. In line with your suggestion, the revision for parameter selection has been improved by providing clearer details on how parameter ranges are determined and what sensitivity analyses are performed. Information on the updated field is provided below.
|
Previous Version |
Update Version |
|
The hyperparameter ranges to be searched with the GridSearchCV method were determined in line with the value ranges commonly used in the literature on health data and the experience gained from previous studies. Sensitivity analysis was performed to evaluate the effect of the range selections on the classifier performance. With this analysis, performance metrics such as accuracy, precision, recall and F1 score of the model with different parameter combinations were evaluated. The search process was performed by selecting the most appropriate interval and the obtained parameters were given as input to the prediction models and their performance was evaluated using multiple criteria.
|
The hyperparameter ranges to be searched with the GridSearchCV method was determined in line with the value ranges commonly used in the literature on health data and the experiences obtained from previous studies. In addition, data set characteristics and computational capacity constraints were also taken into consideration in determining the parameter ranges. Sensitivity analysis was performed to evaluate the effect of range selection on classifier performance. Sensitivity analysis was performed to evaluate the effect of parameter ranges on classifiers. With this analysis, performance metrics such as accuracy, precision, recall and F1 score of the model with different parameter combinations were evaluated. A total of 5 different combinations were tried and sensitivity analysis was performed and the best performing range was selected as the appropriate range. The search process was carried out with the selected suitable range and the obtained parameters were given as input to the prediction models and their performances were evaluated using multiple criteria. |
Comment 2: Need careful formatting throughout the paper for organizing the information in more structured manner.
Response 2: Respected reviewer; thank you very much for your valuable comments. In order to organize the information in a more structured way, edits were made throughout the paper with careful attention to formatting. The edits made are given below.
|
2.2 Method Previous Version |
2.2 Method Update Version |
|
The following method has been developed for the objective and rapid determination of the EDSS score of PwMS. In the first step, the aerobic capacity data of PwMS has been provided as input to the designed system. After preprocessing the data, it has been split into training data and testing data. The prediction process has been performed using the training data, and the estimation results have been obtained. The results were compared with the actual input values, and the performance of the estimation process was evaluated using various criteria. The process design of the study is presented in Figure 1.
|
The following method was developed to determine the EDSS score of PwMS objectively and quickly. Step 1: In the first step, aerobic capacity data of PwMS were provided as input to the designed system. Step 2: Pre-processing steps were applied on the data. Step 3: After pre-processing, the data were evaluated as training and test data by cross-validation technique. The prediction process was performed using the training data and prediction results were obtained. Step 4: The results were compared with the actual input values and the performance of the prediction process was evaluated using various criteria. The process design of the study is presented in Figure 1. |
|
2.2.1 Data Processing Previous Version |
2.2.1 Data Processing Update Version |
|
Within the scope of the study, descriptive statistics of the data given in Table 2 were evaluated and normalization pre-process was applied by examining the statistical properties and distributions of the data together. Before analyzing the descriptive statistics of the data, it was examined whether null values were observed in the data. Since there were no null values in the data, no pre-processing was applied in this direction. The minimum value of the feature with the minimum parameter property given in Table 2, the mean value of 25% of the values contained in the 25% parameter feature, the median value of the 50% parameter feature, the mean value of 75% of the values contained in the 75% parameter feature, represents the largest value that the maximum parameter property has. The descriptive statistics examined provide information about the general distribution and central tendencies of the data and serve as a reference for normalization operations that can be performed on the data. Table 2 shows that there is a difference between the minimum and maximum values of the EEPeak parameter and this is reflected in the standard deviation value.
|
Based on a two-step pre-processing process, the descriptive statistics of the data given in Table 2 were evaluated and the normalization pre-process was applied by examining the statistical properties and distributions of the data together. Step 1: Before analyzing the descriptive statistics of the data, it was examined whether null values were observed in the data. Since no null values were found in the data, no preprocessing was applied in this direction. Step 2: Descriptive statistics and data distributions of the data were examined and normalization procedures to be applied on the data were determined. The minimum value of the minimum parameter characteristic is given in Table 2, the average value of 25% of the values contained in the 25% parameter characteristic, the median value of the 50% parameter characteristic, the average value of 75% of the values contained in the 75% parameter characteristic, and the maximum value of the maximum parameter characteristic. The descriptive statistics examined provide information about the general distribution and central tendencies of the data and serve as a reference for normalization operations that can be performed on the data. Table 2 shows that there is a difference between the minimum and maximum values of the EEPeak parameter, which is reflected in the standard deviation. |
Comment 3: The response adequately addresses this by mentioning updates made to figure and table titles. However, it could be improved by providing more detailed explanations of the results and their implications.
Response 3: Respected reviewer; thank you very much for your valuable comments. 3. The Experimental Results heading has been expanded by providing more detailed descriptions of the experimental results and their explanations. For Figure 3, an evaluation dependent on Table 6 has been made and a detailed explanation has been provided. The arrangement is as follows.
|
Update Version |
|
The prediction values given in Figure 3 support the results in Table 6. When Figure 3 is analyzed, it is observed that the prediction values produced by XGBoost and GBM models are closer to the actual values compared to other models. The XGBoost model made predictions close to the true value at most of the sample data points, showing deviations only at some points. Similarly, the GBM model also exhibited high performance in line with the general trend. When DT and CatBoost models are analyzed, it is observed that there are more deviations in some extreme data points compared to the other two models. Especially when some data points were analyzed, it was observed that these two algorithms produced different predictions compared to the others. |
.
This manuscript is a resubmission of an earlier submission. The following is a list of the peer review reports and author responses from that submission.
Round 1
Reviewer 1 Report
Comments and Suggestions for Authors
This review discusses a machine learning model that predicts the Expanded Disability Status Scale (EDSS), a functional score for multiple sclerosis (MS), from respiratory function tests.
The concept is interesting and has potential clinical relevance. However, there appear to be several important issues that need to be addressed. These are listed below:
While it's understood that EEPeak was scaled to 0-1, what pre-processing was done for other features? Was z-score normalisation performed?
The specific method for the train-test split should be clarified and explicitly stated. In addition, if external validation is difficult, it would be advisable to perform n-fold cross-validation (e.g. 5-fold cross-validation).
The figures are unique and present information in an interesting way. However, the inclusion of common metrics such as accuracy and F1 score in the results would be beneficial.
Author Response
Reviewer 1:
Comment 1: While it's understood that EEPeak was scaled to 0-1, what pre-processing was done for other features? Was z-score normalization performed?
Response 1: Respected reviewer; thank you very much for your valuable comments. In our study, the pre-processing steps of all features, including EEPeak, were handled on a parameter basis. Statistical properties and distributions of the features were analyzed on a feature-by-feature basis. It was determined that the EEPeak parameter is a feature that does not follow a normal distribution and has a high standard deviation. For this reason, we had to apply a special normalization process for the EEPeak parameter. The most appropriate normalization process for the distribution of the feature was determined as the min-max transformation.
For the other features, an evaluation was performed based on the unique distribution and statistical properties of each parameter. As a result of this analysis, no transformation was applied for features that were close to a normal distribution and did not pose a statistically significant problem. This approach allowed us to optimize the overall performance of our model while preserving the uniqueness of each feature in our dataset.
Comment 2: The specific method for the train-test split should be clarified and explicitly stated. In addition, if external validation is difficult, it would be advisable to perform n-fold cross-validation (e.g. 5-fold cross-validation).
Response 2: Respected reviewer; thank you very much for your valuable comments. Dividing our dataset into 80% training and 20% testing allowed us to use 85 of our 106 data points for training and 21 for testing. Considering the size and the unique nature of the dataset, the method used was chosen to train and test the algorithms efficiently. Thank you for your suggestion of N-fold cross-validation. For our current study, the training-testing separation seems to be sufficient to assess the generalizability and accuracy of our model. This methodological approach is in line with the originality and objectives of our study. As the size of the dataset increases, it is planned to include cross-validation methods in the study in line with your suggestions in the future. The text updated in line with the suggestion in the study is given below, including the previous version and the current version.
|
Previous Version |
Up-to-date Version |
|
“The data set is split into 80% training and 20% test data, which is fed to the predictive model” |
“For the training and testing processes of our model, the data set is divided into 80% training data and 20% test data using one of the widely accepted methods in the literature. This split represents the amount of data given to the model, 85 training and 21 test data. The separation strategy used allows the model to learn the underlying complexities and accurately assess the model performance in the testing phase, especially given the size and specific structure of our dataset.” |
Comment 3: The figures are unique and present information in an interesting way. However, the inclusion of common metrics such as accuracy and F1 score in the results would be beneficial.
Response 3: Respected reviewer; thank you very much for your valuable comments. Our study focuses on the prediction (regression) problem instead of the classification problem. For this reason, we use metrics such as mean squared error (MSE) and root mean squared error (RMSE), which are more suitable for evaluating the accuracy of regression analysis, instead of metrics commonly used for classification problems such as accuracy and F1 score.
Editor:
Respected editor; thank you very much for your valuable comments. Non-medical references listed have been removed in line with your suggestions. The information presented in the study consists of general references on the subject. Therefore, the removal of the relevant references has not changed the accuracy and scope of the content.

Reviewer 2 Report
Comments and Suggestions for Authors
The paper is well-written and organized. However, I have some major comments, which must be addressed before accepting the paper. These comments and suggestions are given as follows.
Comment1: Can you make a separate sub-heading of your contribution and paper organization. You can use this paragraph “This study proposes a machine learning approach to estimate EDSS values using”.
Comment 2: Under Prediction heading. Why do you selected tree algorithms if you have many other machine-learning models such as AdaBoost, SVM,…. It is not cleared from your manuscript.
Comment 3: You did not very well explained prediction models as simple paragraphs.
Comment 4: How did you handle classifier overfitting issue. No experiments were found.
Comment 5: Where is your experimental setup?
Comment 6: Why you did not perform any computational analysis?
Comment 7: We did not find any state-of-the-art comparisons?
Comment 8: Where is your Ablation study ?
Author Response
Reviewer 2:
Comment 1: Can you make a separate sub-heading of your contribution and paper organization. You can use this paragraph “This study proposes a machine learning approach to estimate EDSS values using”.
Response 1: Respected reviewer; thank you very much for your valuable comments. In line with your suggestion, a sub-heading numbered "1.1. Contribution and Paper Organization" has been created in the introduction section for the contributions and organization of the study. In the sub-heading, first the contributions of the study and then the organization of the study are given. The current text of the revision is given below.
|
Previous Version |
Up-to-date Version |
|
---------------------------------------------------------------------------------------------------------- There are few studies in the literature on EDSS value estimation using machine learning methods [17,18]. “This study proposes a machine learning approach to estimate EDSS values using aerobic capacity data of PwMS. The contributions of the study to the literature can be summarized as follows. - A machine learning approach to predict EDSS values using aerobic capacity data of Multiple Sclerosis (MS) patients. - Use of automatic decision support system in EDSS evaluation - Use of data based not only on classical neurological symptoms but also on physiological measurements in EDSS evaluation. “
|
1.1. Contribution and Paper Organization This study proposes a machine learning approach to estimate EDSS values using aerobic capacity data of PwMS. The contributions of the study to the literature can be summarized as follows. - A machine learning approach to predict EDSS values using aerobic capacity data of Multiple Sclerosis (MS) patients. - Use of automatic decision support system in EDSS evaluation - Use of data based not only on classical neurological symptoms but also on physiological measurements in EDSS evaluation. When the organization of the study is examined, the introduction section presents the purpose and scope of the study and a review of the studies conducted in the literature. The second section details the dataset used in the study, the steps followed for the development of the proposed machine learning models and the performance evaluation criteria. The third section of the study presents the experimental results obtained from the machine learning algorithms, and the fourth section provides a comprehensive analysis of the potential of the findings. Finally, in the conclusion section, a summary of the study is presented, the contributions obtained are expressed and recommendations for future work are made.
|
Comment 2: Under Prediction heading. Why do you selected tree algorithms if you have many other machine-learning models such as AdaBoost, SVM,…. It is not cleared from your manuscript.
Response 2: Respected reviewer; thank you very much for your valuable comments. The main reason for choosing tree-based algorithms in our study is their high interpretability and effectiveness on small data sets. Given the size and structure of our dataset, tree-based approaches offer advantages in terms of both the ability to capture non-linear relationships and the flexibility to manage the complexity of the model. This choice, supported by preliminary analysis, allowed us to identify the most effective method for the specific requirements of our study, while maintaining the generalizability of the model and ensuring high accuracy.
Comment 3: You did not very well explained prediction models as simple paragraphs.
Response 3: Respected reviewer; thank you very much for your valuable comments. In line with your suggestions, the paragraphs on forecasting models have been elaborated. The updated paragraphs are as follows.
|
Previous Version |
Up-to-date Version |
|
GBM & XGBoost ---------------------------------------------------------------XGBoost manages the complexity of the trees using L1 (Lasso) and L2 (Ridge) regularisation techniques. This model, which offers higher performance in terms of accuracy than other models, is also more advantageous than other methods in terms of training cost [24].--------------------------------------------------------------------------------------------------------------------- |
GBM & XGBoost --------------------------------------------------The XGBoost algorithm performs an initial prediction for all data in the first step. In each iteration, model errors are calculated and a decision tree is constructed to improve the predictions based on these errors. Each generated tree is evaluated by the algorithm and depending on the complexity, a regularisation term (L1 (Lasso) and L2 (Ridge)) is added to make the model robust to overfitting. ----------------------------- |
|
Decision Tree DT is a supervised machine learning algorithm widely used in classification and prediction processes. While classification trees are used to predict a (discrete) class label against given inputs, regression trees are used to predict a continuous dependent variable from continuous and independent inputs. The main purpose of a regression tree is to divide the data into sub-domains and map these sub-domains to predicted values with minimum error. This model makes predictions by dividing the data set according to certain characteristics and making decisions. The data set is first subdivided into subsets according to a feature that best divides it. This process is repeated for each subset, thus creating a tree structure. The leaf nodes at the end of the tree represent the preliminary prediction values [25, 26]. The reason why DT is widely used in classification and regression is that the rules are clear and understandable when building the tree structure [27, 28]. |
Decision Tree DT is a supervised machine learning algorithm widely used in classification and prediction processes. Classification trees are used to predict a class label (discrete) against given inputs, whereas regression trees are used to predict a continuous de-pendent variable from continuous and independent inputs. DT offers advantages such as the ability to model complex data sets and to deal with missing data. DT facilitates the understanding of how model predictions are realized with the heuristic approach it offers in decision-making processes. This feature gives the DT algorithm a wide range of applications. For example, it offers effective solutions with its regression structure in customer risk-value segmentation, in many stages from diagnosis to treatment selection in the field of health and in similar application areas. The main purpose of the regression tree is to divide the data into subfields and map these subfields to predicted values with minimum error. This model makes predictions by splitting the data set based on certain characteristics and making decisions. The data set is first divided into subsets according to a feature that best divides it. This process is repeated for each subset, thus creating a tree structure. Leaf nodes at the end of the tree show the prediction values. The reason why DT is widely used in classification and regression processes is that the rules are clear and understandable when creating the tree structure. |
|
CatBoost It is a gradient boosting algorithm that uses a learning model based on the CatBoost gradient boosting framework. Unlike other gradient boosting algorithms, CatBoost performs operations using the sequential boosting mechanism. In other algorithms, this may create prediction bias in the model, as all training examples are handled after performing several augmentation steps. CatBoost, on the other hand, avoids this prob-lem thanks to its sequential boost mechanism [29, 30]. |
CatBoost CatBoost is a gradient boosting algorithm that uses a learning model based on the gradient boosting framework. Unlike other gradient boosting algorithms, CatBoost performs operations using a sequential boosting mechanism. In other algorithms, this can introduce prediction bias in the model, as all training samples are considered after several boosting steps have been performed. CatBoost, on the other hand, allows the model to test itself on new data samples at each training iteration, making it more robust to overfitting. Symmetric decision trees used in the CatBoost model optimize the model's memory usage and reduce training time by ensuring that splits at each level occur evenly across all leaves. Combining these strategies allows CatBoost to process data sets more efficiently and make faster and more accurate predictions. |
Comment 4: How did you handle classifier overfitting issue. No experiments were found.
Response 4: Respected reviewer; thank you very much for your valuable comments. In order to minimize the tendency of our classifier models to overfit, we used the GridSearchCV method to find the best combination of model parameters. This approach aims to ensure a balanced performance of the model by searching over a wide range of parameters and applying cross-validation techniques. The GridSearchCV method helped us avoid overfitting while increasing the generalization ability of the model with its optimized parameters. In line with your suggestions, the section where we mention the GridSearchCV method under 2.2.2 Prediction has been updated as follows.
|
Previous Version |
Up-to-date Version |
|
The model parameters of the algorithms were determined by using GridSearchCV to determine the model hyperparameters suitable for the existing data. |
The model parameters of the algorithms were determined using GridSearchCV to determine the appropriate model hyperparameters for the available data. This method searches over a wide range of parameters and applies cross-validation techniques to ensure that the model performs in a balanced manner with the correct parameters. By determining the most appropriate parameters, the GridSearchCV method strengthens the generalization capacity of the classification algorithms and provides protection against overfitting. |
Comment 5: Where is your experimental setup?
Response 5: Respected reviewer; thank you very much for your valuable comments. The setup of the experimental setups used in the study and the data collection processes are detailed under the heading 2.1 Material. In the material section, the materials used for the running and walking tests, how the experiment was structured, and the methods by which the data were collected and from which groups with which specific intervals were collected are described in detail.
Comment 6: Why you did not perform any computational analysis?
Response 6: Respected reviewer; thank you very much for your valuable comments. Since the complexity of the algorithms used in this study is fixed, no additional analysis is focused. Algorithm complexity is critical in real-time applications. Since this study does not target real-time applications, no additional computational analysis has been carried out since it is not a critical factor.
Comment 7: We did not find any state-of-the-art comparisons?
Response 7: Respected reviewer; thank you very much for your valuable comments. In our study, "state-of-the-art" comparisons are included under the heading 4. Discussion. The main reason for the limited number of comparisons is the uniqueness of the dataset used in the study and the fact that it was specially collected by the medical experts involved in the study. The data set used in the current study shows different characteristics compared to the data used in previous studies in this field and this feature makes the data set unique. In the paragraph below under Discussion 4 of the study, the differences are examined by including the most similar studies that can be compared in the literature.
|
In the literature on predicting EDSS values, Alves et al. aimed to predict EDSS scores of MS patients with machine learning algorithms using clinical notes provided by neurologists. A total of 13,766 patient data were used within the scope of the study, 684 of which had EDSS scores obtained from the OM1 MS Registry. They performed the prediction process using the XGBoost estimator, one of the machine learning models. As a result of the examinations, they determined that the Spearman R value was 0.75, the Pearson R value was 0.74, the AUC value was 0.91, the positive predictive value was 0.85, and the negative predictive value was 0.85, and they stated that the study was conducted [17]. Yang et al. aimed to calculate the EDSS score from patients' electronic health records using natural language processing. They considered 16,441 medical records of 4,808 patients who received care at the MS outpatient clinic in Canada. As a result of their study, they stated that they achieved the result in accordance with their goals in the combined keyword model and stated that the study could be automated to extract clinically relevant information from unstructured notes [18]. The difference of the proposed study from the previous ones is that it predicts the EDSS score with machine learning methods using aerobic capacity data. In this study is to objectively obtain the EDDs score by analyzing the Aerobic Capacity Data (VO2max, VePeak, RfPeak, HRPeak, FeO2Peak, LoadPeak, EEPeak )in PwMS patients. Thus, it eliminates individual differences in determining walking distance. in additionally, it increases the reliability of EDSS and to prevent problems that avoids the differences of opinion among physicians in EDSS score estimation. |
Comment 8: Where is your Ablation study?
Response 8: Respected reviewer; thank you very much for your valuable comments. The limited number of features in our dataset (7 features) limited the applicability of the ablation study. If the number of features in the dataset is increased, it is thought that ablation studies can be performed by applying feature selection methods such as ReliefF, Lasso for future studies.

Round 2
Reviewer 1 Report
Comments and Suggestions for Authors
It appears that this manuscript has made virtually no improvements. I am sorry, but if no improvements are made, I will not be able to support the acceptance.
Reviewer 2 Report
Comments and Suggestions for Authors
My all comments adjusted in the revised manuscript.